# Management and prognostic prediction of pyogenic liver abscess in a Chinese tertiary hospital: Percutaneous needle aspiration vs catheter drainage

Shumeng Zhang[1], Qiaomai Xu[2], Changhong Liu[3], Zhengjie Wu[2], Zhuoling Chen[4], Silan Gu[2]*

1 Department of Respiratory Disease, Thoracic Disease Center, The First Affiliated Hospital, Zhejiang University School of Medicine, Hangzhou, China, 2 Department of Infectious Diseases, State Key Laboratory for Diagnosis and Treatment of Infectious Diseases, National Clinical Research Center for Infectious Diseases, Collaborative Innovation Center for Diagnosis and Treatment of Infectious Diseases, The First Affiliated Hospital, Zhejiang University School of Medicine, Hangzhou, China, 3 Department of Hepatology, The Fifth People's Hospital of Ganzhou, Ganzhou Institute of Hepatology, Ganzhou, China, 4 Hangzhou Medical College, Hangzhou, China

☉ These authors contributed equally to this work.
* gusilan@zju.edu.cn

**Data Availability Statement:** All relevant data are within the manuscript and its Supporting Information files.

## Abstract

Pyogenic liver abscess (PLA) is a serious infectious disease with high mortality. The aim of our study was to compare the efficacy of percutaneous needle aspiration (PNA) and percutaneous catheter drainage (PCD) for PLA and to assess risk factors for unfavorable prognosis. This retrospective study was performed between 2017 to 2019 in a Chinese tertiary care hospital. We compared the therapeutic effectiveness of PNA versus PCD for PLA and analyzed the risk factors of treatment failure in PLA patients using multivariate logistic regression. A total of 445 patients with PLA were enrolled. The ultrasound-guided percutaneous treatment showed good therapeutic effects on PLA, with a total primary cure rate of 90.1%. PNA appeared to have advantages over PCD, with higher success rates, lower costs, and shorter hospital stays, as well as fewer puncture-induced pain, especially in patients with abscesses of 5–10 cm in diameter. The presence of positive blood culture (OR: 3.32, $p = 0.002$), liver cirrhosis (OR: 3.31, $p = 0.023$), and the length of fever resolution (OR: 1.043, $p = 0.001$) were independent predictors of primary treatment failure. PNA is more advantageous than PCD and is worth considering as a first-line treatment.

## 1. Introduction

Pyogenic liver abscess (PLA) is a global health threat that represents 13% of all intra-abdominal abscesses [1]. A dual blood supply is provided to the liver by the hepatic artery and portal vein, and the liver is connected to the intestine through the bile duct, making it susceptible to bacterial infections that can lead to liver abscesses [2]. With the application of broad-spectrum

**Funding:** This study was supported by National Key R&D Program of China (2023YFC2506000, 2023YFC2506002) and National Key Research and Development Program of China (2021YFC2301805). The funders had no role in study design, data collection and analysis, decision to publish, or preparation of the manuscript.

antibiotics and advances in interventional radiology in recent years, early diagnosis and effective treatment has substantially improved the prognosis of PLA, resulting in a decrease in mortality from 70 to 6–14%; however, its incidence rate continues to rise [3]. Ultrasound-guided percutaneous intervention is an important advancement in PLA treatment and has replaced surgery as the preferred treatment for PLA [4]. Percutaneous needle aspiration (PNA) and catheter drainage (PCD), which are both safe and effective and have a success rate of 70–100%, have also significantly reduced the need for surgical intervention [3]. PNA and PCD are currently considered first-line treatment; however, it is not clear which of the two is more advantageous or which is the better choice for PLA [5]. Furthermore, thorough studies that specifically address the risk factors for primary treatment failure should be completed.

To gain insight into PLA in our hospital, the aim of the current research was to assess epidemiological data, clinical features, radiological manifestations, bacterial characteristics, and treatment and prognosis in our PLA patient cohort. In addition, we compared PNA and PCD in the treatment of PLA to assess awareness of this disease among clinicians.

## 2. Materials and methods

### 2.1 Study population

This retrospective study was conducted at the First Affiliated Hospital, Zhejiang University School of Medicine in Hangzhou, Zhejiang, China. Adult patients with confirmed diagnosis of PLA who had undergone PNA or PCD were enrolled from January 1, 2017, through December 30, 2019. Only the first hospitalization of each patient was considered. The diagnosis of PLA was confirmed with clinical characteristics, radiological manifestations, and bacteriological findings from pus or blood cultures. Fungal or amoebic liver abscesses were excluded. Once diagnosed, PNA or PCD would be performed promptly. The definition of a clinical cure was confirmed upon the relief of clinical symptoms in addition to significant radiological improvement 28 days upon completion of therapy [6]. If the patient's treatment was not classified as curative, it was considered a primary treatment failure. This study was conducted in accordance with the Declaration of Helsinki and approved by the Clinical Research Ethics Committee of the First Affiliated Hospital, Zhejiang University School of Medicine (No. 2023–0115). Patient consent was waived by the Clinical Research Ethics Committee of the First Affiliated Hospital, Zhejiang University School of Medicine due to the retrospective nature of the study and the anonymization of data. The authors had access to information that could identify individual participants during or after data collection. Research data were accessed on March 31, 2023.

### 2.2 Data collection

Using the hospital database, the following data were collected on all patients with PLA: age, sex, comorbidities, prior surgery, radiological manifestations, microbiological findings, laboratory parameters, management, and prognosis. All patients were evaluated based on the Charlson comorbidity index.

### 2.3 Percutaneous needle aspiration

Evacuation of pus from each cavity was performed with the 18-gauge disposable trocar needle. Ultrasound examination is performed every three days to evaluate the size of the abscess, and aspiration was repeated if the abscess cavity did not significantly reduce. Treatment was considered a failure if there was no significant improvement after three aspirations. These patients were then treated with PCD or surgery.

## 2.4 Percutaneous catheter drainage

An 8F pigtail catheter was inserted through the skin into the cavity under ultrasound guidance, and the entry was confirmed by aspiration of pus followed by continuous external drainage. The daily output of the catheter was measured and periodically flushed with saline to prevent occlusion. Repeat sonography was performed on third day to assess size and residual contents, the catheter was removed if clinical signs and ultrasound findings improve. Treatment is considered a failure if PCD is ineffective and surgical treatment is required.

## 2.5 Microbiological data

Blood cultures were incubated at 37˚C for seven days for anaerobic and aerobic cultures (bio-Merieux, France). Pus from PLA was plated on blood agar plates and incubated aerobically and anaerobically at 37˚C for five days. Following incubation, Matrix-assisted laser desorption-ionization-time of flight mass spectrometry was used to determine bacterial strains (Bruker, France).

## 2.6 Statistical analysis

We expressed continuous variables as mean ± standard deviation (SD) or median + interquartile range (IQR), and categorical variables as number (percentage). Differences in means and medians were compared using Mann-Whitney's U-tests and Student t-tests. We compared categorical variables with Chi-square tests. Multivariate logistic regression was performed to identify risk factors related to treatment failure in patients with PLA. Two-sided $p$ values $< 0.05$ were defined as statistically significant. Statistical analysis was conducted using SPSS (version 23.0; SPSS Inc., USA).

# 3. Results

## 3.1 Patient characteristics

A total of 445 PLA patients who had undergone PNA or PCD were hospitalized at the time of the study, with an overall primary cure rate of 90.1%. The baseline demographic and clinical features were summarized in Table 1. Patients ranged in age from 18 to 95 (mean age: 58.3), with a male to female ratio of 2:1. Patients had various underlying comorbidities, with diabetes mellitus (36.6%) and hypertension (30.6%) the most prevalent, followed by tumors (17.3%) and cholelithiasis (18.4%). In total, 150 (33.7%) patients had a history of chronic alcohol use and 39 patients (5.4%) had fatty liver. In addition, 58 (13.0%) patients had a history of surgery within the last 3 months and the median Charlson comorbidity index score was 1. 428 (96.2%) patients had fever, and the median C-reactive protein was 111 mg/L. In total, 11 patients died during hospitalization, accounting for 2.5% of the PLA population. The median hospitalization time was 13 days, and the median hospitalization cost was 23541.7 yuan.

Of these patients, 140 (31.5%) had undergone PNA, and 305 (68.5%) patients had PCD. These two groups were similar in mean age and gender distribution (P > 0.05). The PNA group had a higher incidence of fatty liver ($P < 0.05$). The PCD group had greater prevalence of Malignant tumor and had a higher Charlson comorbidity index ($P < 0.05$). The PCD group had higher levels of aspartate transaminase and C-reactive protein, but lower levels of albumin ($P < 0.05$). The two groups had no significant differences in leukocyte count, alanine transaminase, bilirubin, and serum creatinine.

**Table 1. Demographic and clinical characteristics of 445 patients with pyogenic liver abscess.**

| Patients characteristics | Total (n = 445) | PNA (n = 140) | PCD (n = 305) | P value |
|---|---|---|---|---|
| Age (years) | 58.3 ± 14.0 | 56.5 ± 14.6 | 59.2 ± 12.8 | 0.059 |
| Married | 417 (93.7) | 133 (95.0) | 284 (93.1) | 0.946 |
| Gender Male: Female | 303:142 | 99:41 | 204:101 | 0.422 |
| Area Rural: Urban | 249:196 | 65:75 | 174:131 | 0.494 |
| Smoke | 149 (33.5) | 43 (30.7) | 106 (34.8) | 0.403 |
| Alcohol intake | 150 (33.7) | 46 (32.9) | 104 (34.1) | 0.798 |
| Leukocyte ($10^9$/L) | 10.0 (7.5–13.5) | 9.7 (7.6–12.3) | 10.1 (7.5–14.0) | 0.152 |
| C-reactive protein (mg/L) | 111 (60.7–168.2) | 93.9 (42.9–152.9) | 118 (68.6–178.6) | 0.001 |
| Alanine transaminase (u/L) | 39 (22–66) | 36 (19–57) | 40 (22.5–70.5) | 0.050 |
| Aspartate transaminase (u/L) | 28 (20–47) | 25 (19–39) | 31 (21–50) | 0.020 |
| Bilirubin mg/dl | 11.8 (8.0–19.4) | 10 (6.9–18.0) | 12 (8.1–20.0) | 0.448 |
| Albumin (g/dL) | 31.7 (27.9–35.3) | 32.8 (29.6–36.5) | 31.1 (27.1–34.5) | 0.000 |
| Alkaline phosphatase (u/L) | 139 (99–202) | 126 (92–178) | 144 (105.5–215) | 0.153 |
| Serum creatinine (μmol/L) | 63 (52–77) | 66 (53–78) | 62 (52–75) | 0.390 |
| Charlson comorbidity index score | 1 (0–1) | 0 (0–1) | 1 (0–2) | 0.030 |
| Diabetes mellitus | 163 (36.6) | 51 (36.4) | 112 (36.7) | 0.953 |
| Hypertension | 136 (30.6) | 49 (35.0) | 87 (62.1) | 0.169 |
| Malignant tumor | 77 (17.3) | 16 (11.4) | 61 (20.0) | 0.028 |
| Fatty liver | 39 (8.8) | 23 (16.4) | 16 (5.2) | 0.000 |
| HbsAg | 37 (8.3) | 10 (7.1) | 27 (8.9) | 0.545 |
| Cholelithiasis | 82 (18.4) | 19 (13.6) | 63 (20.7) | 0.074 |
| Liver cirrhosis | 24 (5.4) | 4 (2.9) | 20 (6.6) | 0.109 |
| Previous surgery | 58 (13.0) | 14 (10.0) | 44 (14.4) | 0.199 |
| Fever | 428 (96.2) | 134 (95.7) | 294 (96.4) | 0.729 |
| Hospital stay, days | 13 (9–19) | 12 (8–15) | 14 (9–20) | 0.018 |
| No. of treatment failure | 44 (9.9) | 8 (5.7) | 36 (11.8) | 0.046 |
| No. of patients requiring subsequent surgery | 4 (9.0) | 3 (2.1) | 1 (0.3) | 0.060 |
| Death | 11 (2.5) | 3 (2.1) | 8 (2.6) | 0.763 |
| Hospitalization expenses (yuan) | 23541.7 (16523.3–35128.9) | 17920.2 (13479.2–28252.2) | 25004.8 (18128.6–37782.5) | 0.026 |

## 3.2 Liver abscess characteristics

All the patients had accepted abdominal ultrasonography, and the mean maximum abscess diameter was 7.2 ± 2.5 cm. The size of most of PLAs (66.3%) was 5–10 cm, and 19.8% were less than 5cm, with the remaining 13.9% greater than 10 cm. Most cases of PLA were localized to right lobe of liver (67.6%), and 22.0% were confined to the left lobe, with the remaining 10.3% of cases involving both lobes. In this study, 50.1% of abscesses were single abscesses, and multiple abscesses were found in 49.9% of patients (Table 2).

## 3.3 Microbiological characteristics

We observed blood or pus cultures in all cases. We found 279 cases (279/445, 62.7%) with positive pus-culture microbial reports and 52 cases (52/445, 11.7%) with positive blood cultures. The most common organism in pus-culture was *Klebsiella pneumonia* (207/279, 74.2%) followed by *Escherichia coli* (14/279, 5.0%) and *Streptococcus anginosus* group (12/279, 4.3%). Similarly, the most common organism in blood culture was *Klebsiella pneumonia* (31/52, 59.6%) followed by *Escherichia coli* (5/52, 9.6%) and Staphylococcus species (5/52, 9.62%) (Table 3).

**Table 2. Image characteristics of 445 patients with pyogenic liver abscess.**

| Characteristics | PNA (n = 140) | PCD (n = 305) | Total (n = 445) |
|---|---|---|---|
| Number | | | |
| Single | 78 (55.7) | 145 (47.5) | 223 (50.1) |
| Multiple | 62 (44.3) | 160 (52.5) | 222 (49.9) |
| Site | | | |
| Right | 98 (70.0) | 203 (66.7) | 301 (67.6) |
| Left | 29 (20.7) | 69 (22.6) | 98 (22.0) |
| Both | 13 (9.3) | 33 (10.8) | 46 (10.3) |
| Size | | | |
| Mean diameter (cm) | 6.4 ± 2.6 | 7.5 ± 2.4 | 7.2 ± 2.5 |
| < 5cm | 42 (30.0) | 46 (15.1) | 88 (19.8) |
| 5 ≤ diameter <10cm | 84 (60.0) | 211 (69.2) | 295 (66.3) |
| ≥ 10cm | 14 (10.0) | 48 (15.7) | 62 (13.9) |

PNA, percutaneous needle aspiration; PCD, percutaneous catheter drainage.

## 3.4 Comparison of treatment and outcome between PNA and PCD

All patients received intravenous antibiotic therapy, and the antibiotic therapy was adjusted based on the results of culture and sensitivity test of pus. There were 3 deaths in the PNA group and 8 deaths in the PCD group, with no significant difference in mortality between

**Table 3. Microbiological data of 445 patients with pyogenic liver abscess.**

| Characteristics | PNA | PCD | Total, n (%) |
|---|---|---|---|
| Organism cultured (pus) | n = 96 | n = 183 | n = 279 |
| 1.*Klebsiella pneumonia* | 79 (82.3) | 128 (70.0) | 207 (74.2) |
| 2. *Escherichia coli* | 3 (3.1) | 11 (6.0) | 14 (5.0) |
| 3. *Streptococcus anginosus group* | 4 (4.2) | 8 (4.4) | 12 (4.3) |
| 4. *Staphylococcus species* | 2 (2.1) | 8 (4.4) | 10 (3.6) |
| 5. *Enterococcus faecium* | 0 (0.0) | 7 (3.8) | 7 (2.5) |
| 6. *Enterococcus faecalis* | 0 (0.0) | 5 (2.7) | 5 (1.8) |
| 7. *Enterobacter cloacae* | 2 (2.1) | 1 (0.5) | 3 (1.1) |
| 8. *Pseudomonas aeruginosa* | 1 (1.0) | 1 (0.5) | 2 (0.7) |
| 9. *Klebsiella oxytoca* | 1 (1.0) | 1 (0.5) | 2 (0.7) |
| 10. *Citrobacter koseri* | 0 (0.0) | 2 (1.1) | 2 (0.7) |
| 11. *Enterobacter aerogenes* | 0 (0.0) | 2 (1.1) | 2 (0.7) |
| 12. Other | 4 (4.2) | 9 (4.9) | 13 (4.7) |
| Organism cultured (blood) | n = 9 | n = 43 | n = 52 |
| 1.*Klebsiella pneumonia* | 5 (55.6) | 26 (60.5) | 31 (59.6) |
| 2. *Escherichia coli* | 0 (0.0) | 5 (11.6) | 5 (9.6) |
| 3. *Staphylococcus species* | 1 (11.1) | 4 (9.3) | 5 (9.6) |
| 4. *Pseudomonas aeruginosa* | 1 (11.1) | 1 (2.3) | 2 (3.8) |
| 5. *Enterobacter asburiae* | 1 (11.1) | 1 (2.3) | 2 (3.8) |
| 6. *Streptococcus anginosus group* | 0 (0) | 2 (4.7) | 2 (3.8) |
| 7. *Enterobacter cloacae* | 1 (11.1) | 0 (0.0) | 1 (1.9) |
| 8. *Acinetobacter baumannii* | 0 (0.0) | 1 (2.3) | 1 (1.9) |
| 9. *Blastomyces dermatitidis* | 0 (0.0) | 1 (2.3) | 1 (1.9) |
| 10. *Citrobacter freundii* | 0 (0.0) | 1 (2.3) | 1 (1.9) |
| 11. *Stenotrophomonas maltophilia* | 0 (0.0) | 1 (2.3) | 1 (1.9) |

**Table 4. Outcome comparison between PNA and PCD in 445 patients with pyogenic liver abscess.**

| Patients characteristics | Total (n = 445) | | | diameter < 5cm (n = 88) | | | 5 ≤ diameter <10cm (n = 295) | | | diameter ≥ 10cm (n = 62) | | |
|---|---|---|---|---|---|---|---|---|---|---|---|---|
| | PNA (n = 140) | PCD (n = 305) | P value | PNA (n = 42) | PCD (n = 46) | P value | PNA (n = 84) | PCD (n = 211) | P value | PNA (n = 14) | PCD (n = 48) | P value |
| No. of puncture-induced pain | 52 (37.1) | 149 (48.9) | 0.021 | 15 (35.7) | 21 (45.7) | 0.349 | 31(36.9) | 104 (49.3) | 0.054 | 6 (42.9) | 24 (50.0) | 0.645 |
| Resolution of fever | 4 (2–8) | 6 (3–12) | 0.094 | 4 (1–8) | 13 (8–17) | 0.402 | 4 (2–8) | 6 (3–12) | 0.081 | 7 (5–9) | 8 (3–13) | 0.544 |
| Hospital stay | 12 (8–15) | 14 (9–20) | 0.018 | 11 (8–17) | 5 (2–10) | 0.369 | 10 (7–14) | 13 (9–20) | 0.010 | 16 (14–20) | 17 (11–21) | 0.605 |
| No. of patients requiring subsequent surgery | 3 (2.1) | 1 (0.3) | 0.060 | 0 (0.0) | 0 (0.0) | NA | 3 (3.6) | 1 (0.5) | 0.038 | 0 (0.0) | 0 (0.0) | NA |
| No. of patient death | 3 (2.1) | 8 (2.6) | 0.763 | 1(2.4) | 2 (4.3) | 0.616 | 2 (2.4) | 6 (2.8) | 0.826 | 0 (0.0) | 0 (0.0) | NA |
| No. of treatment failure | 8 (5.7) | 36 (11.8) | 0.046 | 3(7.1) | 6 (13.0) | 0.367 | 5 (6.0) | 24 (11.4) | 0.159 | 0 (0.0) | 6 (12.5) | 0.169 |
| Hospitalization expenses | 17920.26 (13479.2–28252.2) | 25004.81 (18128.6–37782.5) | 0.026 | 17280.0 (11189.1–34932.4) | 24521.7 (18010.3–44519.6) | 0.203 | 17240.0 (13603.37–25713.4) | 24720.0 (17966.7–36265.0) | 0.018 | 25079.4 (20504.6–30439.1) | 25858.7 (19162.5–43089.2) | 0.195 |

NA, not assessable.

groups. The mean abscess size was 6.4 ± 2.6 cm in the PNA group and 7.5 ± 2.4 cm in the PCD group. The mean hospital stay was shorter in the PNA group than in the PCD group (*p* = 0.018), and the PNA group accrued fewer hospitalization expenses as well (*p* = 0.027). The number of puncture-induced pain in the PNA group was also less than in the PCD group, and the patients undergoing PNA had a higher treatment success rate (*p* = 0.046) than those undergoing PCD. Interestingly, when the diameter of the abscess was less than 5 cm or larger than 10 cm, the two groups did not differ; yet if the abscess was 5–10 cm in diameter, the PNA group had a shorter hospital stay (*p* = 0.010) and had lower hospital costs (*p* = 0.018) than the PCD group (Table 4).

### 3.5 Risk factors of treatment failure in PLA patients

Moreover, 44 patients (9.9%) in the study were defined as primary treatment failure, including 11 patients (2.5%) who died during hospitalization and 33 patients (7.4%) who required re-treatment. Among the 44 defined as primary treatment failure, 8 patients were from the group PNA and 36 were from the PCD group. A multivariate analysis of treatment failure risk factors is presented in Table 5, and candidate variables were assessed using a univariate logistic regression in S1 Appendix. According to the multivariate logistic regression, liver cirrhosis (OR: 3.31, 95% CI: 1.176–9.338; *p* = 0.023), positive blood culture (OR: 3.32, 95% CI: 1.558–7.068; *p* = 0.002), and the length of fever resolution (OR: 1.043, 95% CI: 1.02–1.07; *p* = 0.001) were associated significantly with poor prognosis in patients with PLA.

### 4. Discussion

From 2017 to 2019, 445 patients with PLA accepted ultrasound-guided percutaneous treatment in our center. Ultrasound-guided percutaneous treatment showed good therapeutic effects and safety on PLA, with a total primary cure rate of 90.1%.

Most of the patients in our study were male with a mean age of 58.3 ± 14.0 years, which is similar to a previous study that reported a mean age of 59.1 ± 12.7 years [7]. The symptoms of PLA vary with low specificity, including fever, vomiting, nausea, fatigue, and abdominal pain. The majority of patients had elevated C-reactive protein levels and white blood cell counts, as

**Table 5. Risk factors for primary treatment failure in 445 patients with pyogenic liver abscess analyzed by multivariate logistic regression.**

| Variables | Success (n = 401) | Failure (n = 44) | Univariate analysis | | Multivariate analysis | |
|---|---|---|---|---|---|---|
| | | | OR (95% CI) | P value | OR (95% CI) | P value |
| Age (years) | 58.0 ± 13.8 | 61.8 ± 15.4 | 0.98 (0.96–1.003) | 0.080 | | |
| Charlson comorbidity index score | 1 (1–0) | 1 (0–2) | 1.32 (1.04–1.69) | 0.025 | | |
| Liver cirrhosis | 18 (4.5) | 6 (13.6) | 3.36 (1.26–8.97) | 0.016 | 3.31 (1.18–9.34) | 0.023 |
| Previous surgery | 47 (11.7) | 11 (25.0) | 2.51 (1.19–5.30) | 0.016 | | |
| Aspartate transaminase (u/L) | 28 (20–44) | 33 (23–63) | 1.01 (1.00–1.01) | 0.014 | | |
| Alkaline phosphatase (u/L) | 137 (97–195) | 196 (114–258) | 1.003 (1.001–1.004) | 0.008 | | |
| Organism cultured (blood) | 39 (9.7) | 14 (31.8) | 4.33 (2.12–8.86) | <0.001 | 3.32 (1.56–7.07) | 0.002 |
| ICU admission | 5 (1.2) | 4 (9.1) | 7.92 (2.04–30.69) | 0.003 | | |
| Resolution of fever, days | 5 (2–10) | 9 (5–15) | 1.05 (1.02–1.07) | 0.001 | 1.04 (1.02–1.07) | 0.001 |
| Hospital stay, days | 13 (8–19) | 14 (10–20) | 1.03 (1.01–1.06) | 0.003 | | |

well as decreased blood albumin level, which were comparable with other studies [8]. Biliary diseases act as a major risk factor for PLA in many recent reports from western countries, most often because of biliary infections [9]. Interestingly, among the underlying diseases in our study, diabetes mellitus ranked highest, followed by biliary tract diseases. Diabetes remains a major risk factor for PLA, with a risk rate of 3.6–9-fold higher than the general population [8]. Uncontrolled hyperglycemia may impair the host's defense mechanisms by reducing barrier function, neutrophil chemotaxis, and activation of mononuclear phagocytes in patients with diabetes [10]. Hyperglycemia will also aggravate metabolic disorders and bacterial growth in the gastrointestinal tract and liver, and may even lead to larger abscesses than in nondiabetic patients [11].

Our study showed that 9.9% of patients experienced primary treatment failure even following aggressive treatment. After adjustment of baseline differences by the multivariate logistic regression, independent predictors of primary treatment failure included liver cirrhosis, positive blood culture, and the length of fever resolution. Due to patient's impaired immunity, liver cirrhosis has been identified as a major risk factor for PLA [12].

Data also suggests that gut microbial dysbiosis is common in cirrhotic patients, and the use of antibiotics in these patients may exacerbate the dysbiosis [13], thus a prominent alteration in their gut microbiota may precipitate an oral bacterial species invasion [14]. In addition, spontaneous bacterial peritonitis is a serious but common complication in the liver cirrhosis population, in which ascites could carry a variety of infectious bacteria [15]. These conditions may make patients with liver cirrhosis more susceptible to PLA.

In our cohort, positive blood culture was identified as a risk factor for primary treatment failure, suggesting that sepsis can further lead to the spread of infections in multiple organs throughout the body. This could result in metastatic infections and the further development of invasive liver abscess syndrome, which has a mortality rate of up to 14% [16]. Lung abscesses, endogenous endophthalmitis, central nervous system infections, and necrotizing fasciitis may also occur, which can further increase the cost of hospitalization as well as the mortality in patients with PLA [17]. The length of fever resolution was also associated with primary treatment failure, and poor infection control generally leads to persistent fever. There are many reasons for poor control, including drug resistant bacteria, inadequate drug concentrations, large or multiple infection lesions, and underlying diseases [6]. Therefore, prolonged fever is also an early warning sign for poor clinical outcome.

In our study, most abscesses were located in the right lobe of the liver and ranged from 5–10 cm in diameter, about half of which were single abscesses. They may have been located

here because the right hepatic lobe receives the most blood from the portal veins because of its large size [8]. There is evidence that hematogenous seeding in PLA occurs primarily through the portal systems [10]. In general, abscesses should be treated by removing the pus as soon as possible and by selecting the appropriate antibiotic according to drug sensitivity.

Interestingly, in some previous research, both PCD and PNA have been shown to be effective in treating patients with PLA; however, in other studies, PCD appears more effective than PNA [18,19]. In our study, PNA had advantages over PCD, including higher success rates, lower costs, shorter hospital stays, and fewer puncture-induced pain. In particular, in patients with 5–10 cm abscesses, PNA significantly reduced hospital stays, hospital expenses, and the risk of requiring further surgical interventions. In the remaining patients with abscesses smaller than 5 cm or larger than 10 cm, the two treatment options did not differ significantly. Although this finding conflicts with some previous research, it may be related to the insufficiently strict inclusion of abscess types in previous studies, including amoebic, pyogenic, mixed, and indeterminate abscesses [18,19].

PNA is worth considering as a first-line treatment for PLA as it offers more advantages over PCD. First, compared with PCD, PNA is simple to operate and can rapidly drain all pus by negative pressure, which is extremely effective in the removal of pathogens [3,20]. Second, due to the smaller caliber of the 18G metallic needle compared to the 8F catheter, there is less trauma to the liver. Third, in multiple or multi-loculated abscesses, the 18G metallic needle is easier to manipulate and can pass through the septae between locules of the abscess. Furthermore, PNA is more easily accepted by patients in terms of comfort as it does not require a tube to be left in the body, and there is no risk of lumen pulling pain, dislodgement, fracture, or leakage [3]. Finally, the costs of PNA is cheaper than PCD. In addition, the drainage tubes of PCD require regular sterilization and dressing changes, which can increase PCD medical costs. The main disadvantage of PNA is it may require multiple suctions during the treatment of PLA, but PCD also cannot guarantee a single session successful outcome [21]. Some studies had shown that most patients need only one or two sessions of PNA, and repeated aspiration do not significantly increase in morbidity and mortality [3,22]. The findings of this study could help physicians make more informed choices when selecting treatment modalities in their clinical work. However, the optimal PLA treatment remains to be confirmed by further randomized controlled trials.

In addition, identifying pathogens from blood or abscess cultures is extremely important for diagnosing and treating PLA. In our study, the positive rate of microbial culture was lower than that in other studies [23,24]. This low microbiological detection rate may be because many patients received antibiotics in their local hospitals before they arrived at our national clinical medical center, and the early use of antibiotics can quickly reduce microorganisms in the body [25]. *Escherichia coli* is the most common pathogenic bacteria in PLA in western countries, followed by *Klebsiella pneumoniae*, *Enterococcus*, and *Streptococcus* [6]. However, most PLA were caused by *Klebsiella* pneumoniae and *Escherichia coli* in our study, which is often seen in other regions of China [7,10]. *Klebsiella pneumoniae* is also the most common PLA pathogen in Southeast Asia and is currently increasing significantly in the United States [11,26]. Diabetic patients are also more likely to be infected with *Klebsiella pneumoniae*, although the biological mechanism remains of infection is not known [27]. Some studies suggest that the mechanism may be associated with tissue hyperglycemia and the predilection of *Klebsiella pneumoniae* [11]. Infections caused by *Escherichia coli*, in comparison, are more prevalent in PLA patients without diabetes, which may be due to a greater frequency of abdominal surgery and liver infection [8].

There are several limitations to the current study. Because blood samples were usually taken and cultured when the patient's temperature exceeded 38.5°C in our hospital to improve

blood culture positivity, we suspect that the rate of positive blood culture may have been missed in a portion of PLA patients. Therefore, the results may be biased. In addition, the findings from this retrospective single-center study should be verified through multi-center studies. Further, most patients in our study had abscesses ranging from 5–10cm, and the number of patients with abscesses smaller than 5cm or larger than 10cm was minimal. Therefore, a larger cohort is needed to verify the selection of the optimal treatment scheme among these patients.

## 5. Conclusions

In conclusion, the ultrasound-guided percutaneous treatment showed good therapeutic effects on PLA, with a total primary cure rate of 90.1%. Presence of positive blood culture, liver cirrhosis, and the length of fever resolution were independent predictors of primary treatment failure. Based on our experience, PNA has advantages over PCD, with higher success rates, lower costs, shorter hospital stays, and fewer puncture-induced pain, especially in patients with abscesses of 5–10 cm in diameter. However, further assessment is required to validate these findings. With the additional advantages of simpler operation and lower costs, PNA is worth considering as a first-line treatment for PLA.

## Supporting information

**S1 Appendix.**
(DOCX)

## Author Contributions

**Conceptualization:** Shumeng Zhang, Silan Gu.

**Data curation:** Qiaomai Xu, Changhong Liu, Zhengjie Wu, Zhuoling Chen.

**Formal analysis:** Qiaomai Xu.

**Investigation:** Changhong Liu, Zhengjie Wu, Zhuoling Chen.

**Methodology:** Shumeng Zhang.

**Supervision:** Shumeng Zhang, Silan Gu.

**Writing – original draft:** Shumeng Zhang, Qiaomai Xu, Changhong Liu.

**Writing – review & editing:** Silan Gu.

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
