## [Decision Letter · Decision Letter 0]

18 Sep 2024

PONE-D-24-30205Management and prognostic prediction of pyogenic liver abscess in a Chinese tertiary hospital: percutaneous needle aspiration vs catheter drainagePLOS ONE

Dear Dr. Silan,

Thank you for submitting your manuscript to PLOS ONE. After careful consideration, we feel that it has merit but does not fully meet PLOS ONE’s publication criteria as it currently stands. Therefore, we invite you to submit a revised version of the manuscript that addresses the points raised during the review process.

We look forward to receiving your revised manuscript.

Kind regards,

Zhaoqing Du, Ph.D

Academic Editor

PLOS ONE

**Journal Requirements:**

National Key R&D Program of China 2023YFC2506000, 2023YFC2506002

Reviewers' comments:

Reviewer's Responses to Questions

**Comments to the Author**

1. Is the manuscript technically sound, and do the data support the conclusions?

Reviewer #1: Yes

Reviewer #2: Yes

2. Has the statistical analysis been performed appropriately and rigorously? 

Reviewer #1: Yes

Reviewer #2: N/A

3. Have the authors made all data underlying the findings in their manuscript fully available?

Reviewer #1: No

Reviewer #2: Yes

4. Is the manuscript presented in an intelligible fashion and written in standard English?

Reviewer #1: Yes

Reviewer #2: Yes

5. Review Comments to the Author

**Reviewer #1:** 1.Why the diameter of the pigtail tube is 8F, and the 8F tube diameter is small, why do not use 10-12F.

2.What are the inclusion and exclusion criteria for the studies? Patients diagnosed with PLA were included? Did the PLA liquefied very good? Have any multilocular liver abscess?

3.Once diagnosed with a bacterial liver abscess, PNA or PCD occurs quickly, and what about patients with low platelets or those taking anticoagulants? Does PNA enhance the bleeding risk

4.Most of the previous studies show that PCD is better than PNA. Is it because the pigtail tube you use has a small diameter? Or is the drain tube placed in a poor position?

5.Does a larger PLA have only one puncture point for PNA treatment?

6.PCD failure needs surgical treatment, why not undertake PNA treatment after PCD treatment failure?

7.The clinical characteristics of patients with PLA treated with PNA and PCD were not compared, could there be a conclusion bias? Could there be a large difference in the average abscess size?

8.What is the use of antibiotics? Is there any significant difference? Will it affect the study results?

9.Previous studies have shown that PNA is mainly used for the treatment of patients with liver abscess with small diameter. If possible, please show the comparative picture. I am interested in the treatment effect of PNA on larger liver abscesses.

10.What are the surgery-related complications, please specify them.

**Reviewer #2: **This paper has taken an important step in exploring the treatment of liver abscess and has shown potential research value.

However, after careful review, I find that the content of the article is somewhat limited and may not fully demonstrate the depth and breadth of the research. To enhance the persuasiveness and impact of the article, I suggest considering the following aspects for supplementation and improvement:

1. Additional Experimental Content: If conditions permit, I recommend conducting further experiments, particularly those that can directly support or supplement the findings from the current retrospective analysis. This will help solidify the research conclusions and provide readers with more empirical evidence.

2.Multi-center Study: Considering the potential limitations of a single-center study, if data from other institutions can be obtained to conduct a multi-center study, it will greatly enhance the external validity and reliability of the research. This will not only strengthen the generality of the results but also help identify potential differences between different centers, thereby enriching the research content.

6. PLOS authors have the option to publish the peer review history of their article (what does this mean?). If published, this will include your full peer review and any attached files.

Reviewer #1: **Yes: **JinHua Cui

Reviewer #2: No

---

## [Author Response · Author response to Decision Letter 0]

23 Oct 2024

Dear editors and reviewers: 

Thanks a lot for your kind consideration on our manuscript PONE-D-24-30205 and great helps during the process of revision. We also want to express our deep thanks to reviewers for their comments and good suggestions. In particular, we highly appreciated that the reviewers not only had raised some questions to improve the manuscript but also had supplied some perfect solutions, which made this revision more efficient and accurate. 

We take these questions and suggestions very seriously and have tried our best to revise every issue according to the reviewers’ suggestions. All the changes after revision have been highlighted with yellow color of fonts; or you can see all the tracked changes in a related manuscript file with revised yellow markers. Besides, we also tried our best to explain why such changes were done and how these changes affected the results or conclusions of this manuscript before and after revision in details.

Thanks a lot for your kind attention. Please do not hesitate to contact us if there is anything we can do for you. 

The followings are our responses to the reviewers' comments and the list of corrections. 

Sincerely,

Silan Gu

State Key Laboratory for Diagnosis and Treatment of Infectious Diseases, Collaborative Innovation Center for Diagnosis and Treatment of Infectious Diseases, the First Affiliated Hospital, College of Medicine, Zhejiang University, 310003 Hangzhou, China.

Tel/Fax: +86 571 87236459.

E-mail: gusilan@zju.edu.cn

Journal Requirements:

Response: Thank you for your helpful suggestion. We read the requirements carefully and made the necessary changes.

National Key R&D Program of China 2023YFC2506000, 2023YFC2506002

Response: Thank you for your helpful suggestion. We have added a fund and made the following statement in the article and cover letter: this study was supported by National Key R&D Program of China (2023YFC2506000, 2023YFC2506002) and National Key Research and Development Program of China (2021YFC2301805). Theses funders had no role in study design, data collection and analysis, decision to publish, or preparation of the manuscript.

Review Comments to the Author

Reviewer #1: 

1.Why the diameter of the pigtail tube is 8F, and the 8F tube diameter is small, why do not use 10-12F.

Response: Thank you for your helpful suggestion. This is a retrospective study in which we observed that pigtail tubes of 8F size were used in clinical treatment. We believe that it is possible that 1) for most patients with liver abscesses, 8F tubes are sufficient to continuously drain the pus for therapeutic effect, and 2) relatively speaking, tubes of smaller diameter cause less possible injury and provide better comfort. Of course, a small diameter tube does have the potential for blockage, so the tube will be flushed with saline periodically after placement to prevent blockage (Line 89-90). 

Your suggestion of using a pigtail tube with a diameter of 10-12F is indeed a very good one, and we would also like to learn more about what these differences might lead to if a subsequent RCT study is conducted.

2.What are the inclusion and exclusion criteria for the studies? Patients diagnosed with PLA were included? Did the PLA liquefied very good? Have any multilocular liver abscess?

Response: Thank you for your helpful suggestion. The purpose of this study was to compare the differences between PNA and PCD in the treatment of Pyogenic liver abscess. Only adult patients with confirmed diagnosis of PLA who had undergone PNA or PCD were enrolled. And only the first hospitalization of each patient was considered. The diagnosis of PLA was confirmed with clinical characteristics, radiological manifestations, and bacteriological findings from pus or blood cultures. Fungal or amoebic liver abscesses were excluded. (Line 59-64). Once diagnosed, PNA or PCD will be performed immediately in these presence of liquefied liver abscesses (Line 64). And ultrasound examination is performed every three days to evaluate the size of the abscess, further operation options are then made based on the results of the examination. (Line 82-83, Line 90-91) There was a total of 222 patients (222/445, 49.9%) with multiple abscesses (Table 2).

3.Once diagnosed with a bacterial liver abscess, PNA or PCD occurs quickly, and what about patients with low platelets or those taking anticoagulants? Does PNA enhance the bleeding risk

Response: Thank you for your helpful suggestion. Only patients who had undergone PNA or PCD were enrolled (Line 59-60). Patients who were not candidates for puncture or for whom there were clinical contraindications were not included in this study.

4.Most of the previous studies show that PCD is better than PNA. Is it because the pigtail tube you use has a small diameter? Or is the drain tube placed in a poor position?

Response: Thank you for this comment. Our study found that PNA was superior to PCD. As the mean hospital stay was shorter in the PNA group than in the PCD group and the PNA group accrued fewer hospitalization expenses as well (Line 152-154). Generally speaking, an 8F tube can meet the needs of most patients with pyogenic liver abscess in this study, and we regularly flush the tube with saline to ensure smooth drainage. In addition, we regularly perform ultrasound examinations determine whether drainage is appropriate (Line 89-91). We also agree that further research, such as multicenter studies, are needed to confirm this conclusion, and we emphasized this in the limitations section (Line 276-278). 

Our review of the literature indicated that only one randomized controlled trial (RCT) of liver abscess (S.C. Yu et Hepatology 39(4) (2004) 932-8) strictly included PLA, the size of pigtail tube in this RCT and treatment were the same as in our study. The findings of the other two meta-analyses conflict with the findings from this previous RCT and our study. This might be because these other studies included patients with multiple types of abscesses (amoebic, pyogenic, mixed, and indeterminate) (Line 237-239). Further research that focuses on PLA is needed to confirm our conclusions.

5.Does a larger PLA have only one puncture point for PNA treatment?

Response: Thank you for this comment. When performing a PNA, as much pus as can be extracted will be drained off, and sometimes more than one puncture site may be necessary unless the abscess has not completely liquefied, and ultrasound examination is performed every three days to evaluate the size of the abscess, and aspiration was repeated if the abscess cavity did not significantly reduce (Line 82-85).

6.PCD failure needs surgical treatment, why not undertake PNA treatment after PCD treatment failure?

Response: Thank you for this comment. This is a very interesting point because this was a retrospective study and we observed that patients who failed PCD would not try PNA again. This may be related to the experience of the clinicians as well as previous literature including meta-analyses, which showed PCD to be superior to PNA. These reasons may have contributed to the eventual choice made in these cases.

7.The clinical characteristics of patients with PLA treated with PNA and PCD were not compared, could there be a conclusion bias? Could there be a large difference in the average abscess size?

Response: Thank you for this comment. In Table 1 we show the demographic and clinical characteristics of these two groups (PNA and PCD) of patients with pyogenic liver abscesses. These two groups were similar in mean age and gender distribution (P > 0.05). The PNA group had a higher incidence of fatty liver (P < 0.05). The PCD group had greater prevalence of Malignant tumor and had a higher Charlson comorbidity index (P < 0.05). The PCD group had higher levels of aspartate transaminase and C-reactive protein, but lower levels of albumin (P < 0.05). The two groups had no significant differences in leukocyte count, alanine transaminase, bilirubin, and serum creatinine (Line 124-129). 

And in Table 2 we show the imaging characteristics of the abscesses in these two groups of patients. All the patients had accepted abdominal ultrasonography, and the mean maximum abscess diameter was 7.2 ± 2.5 cm. The size of most of PLAs (66.3%) was 5–10 cm, and 19.8% were less than 5cm, with the remaining 13.9% greater than 10 cm. Most cases of PLA were localized to right lobe of liver (67.6%), and 22.0% were confined to the left lobe, with the remaining 10.3% of cases involving both lobes. In this study, 50.1% of abscesses were single abscesses, and multiple abscesses were found in 49.9% of patients (Line 131-136). 

In order to minimize the influence of abscess size on the results, we referred to several published papers and classified the abscesses into three groups: less than 5 cm, 5-10 cm, and more than 10 cm, and then compared the differences in the therapeutic efficacy of PNA and PCD in abscesses of different sizes, respectively (Table 4).

8.What is the use of antibiotics? Is there any significant difference? Will it affect the study results?

Response: Thank you for this comment. All patients received intravenous antibiotic therapy (mostly third-generation cephalosporins), and the antibiotic regimen was adjusted based on the results of pus cultures and drug sensitivity tests. (Line 148-149)

9.Previous studies have shown that PNA is mainly used for the treatment of patients with liver abscess with small diameter. If possible, please show the comparative picture. I am interested in the treatment effect of PNA on larger liver abscesses.

Response: Thank you for this comment. We reviewed a total of 8 reported RCTs comparing the efficacy of PCD and PNA in the treatment of liver abscesses. Of these, a total of 6 studies concluded that PCD was superior to PNA and 2 studies concluded that PNA was superior to PCD. 2 studies defined liver abscesses greater than 10 cm in diameter, and both concluded that PCD was superior to PNA. Of the 8 studies mentioned above, only one study enrollment included only pyogenic liver abscesses (removal of amoebas, fungi, etc.) and concluded that PNA was superior to PCD. However, there are no reported RCTs of pyogenic large liver abscesses. 

The size reports of abscesses in all patients in this study were obtained from the electronic reports of ultrasound examinations. And in table 4, we grouped all cases according to abscess size based on the ultrasound report and found that there was no statistically significant difference between PCD and PNA in pyogenic liver abscesses with a diameter greater than 10 cm. This may be related to the following points: on the one hand, the number of patients with large abscesses is relatively small, and Secondly, in the group of patients with large abscesses with a diameter exceeding 10cm, clinical doctors have a much lower chance of selecting PNA based on their clinical experience compared to abscesses with other diameters. For this reason, we are planning to follow up with a more detailed RCT in the hope of obtaining more information.

10.What are the surgery-related complications, please specify them.

Response: Thank you for this comment. This refers to the pain caused by puncture. We have modified it to ' puncture-induced pain' for greater clarity.

Reviewer #2: This paper has taken an important step in exploring the treatment of liver abscess and has shown potential research value.

However, after careful review, I find that the content of the article is somewhat limited and may not fully demonstrate the depth and breadth of the research. To enhance the persuasiveness and impact of the article, I suggest considering the following aspects for supplementation and improvement:

1. Additional Experimental Content: If conditions permit, I recommend conducting further experiments, particularly those that can directly support or supplement the findings from the current retrospective analysis. This will help solidify the research conclusions and provide readers with more empirical evidence.

Response: Thank you for your helpful suggestion. Indeed, as you say, the findings from this retrospective single-center study should be verified through multi-center studies. Indeed, as you say, the idea of retrospective studies needs to be further substantiated by multicenter studies, and we emphasized this in the limitations section (Lines 276-278). Our study has carried out some work on the controversies existing in previous studies and obtained corresponding conclusions, which support some of the conclusions of previous studies, such as PNA has advantages over PCD, with higher success rates, lower costs, shorter hospital stays, and fewer puncture-induced pain, especially in patients with abscesses of 5–10 cm in diameter. As a next step, we will further design a multicenter, randomized controlled trial based on this study, hoping that it will help consolidate the research findings and provide more empirical evidence for clinical doctors.

2.Multi-center Study: Considering the potential limitations of a single-center study, if data from other institutions can be obtained to conduct a multi-center study, it will greatly enhance the external validity and reliability of the research. This will not only strengthen the generality of the results but also help identify potential differences between different centers, thereby enriching the research content.

Response: Thank you for your helpful suggestion. As you point out, multicenter studies can strengthen the generality of the results but also help identify potential differences between different centers, thereby enriching the research content. For this purpose, we first reviewed the published literature. We have reviewed a total of 8 reported RCTs comparing the efficacy of PCD and PNA in the treatment of liver abscesses. Of these, a total of 6 studies concluded that PCD was superior to PNA and 2 studies concluded that PNA was superior to PCD. Of the 8 studies mentioned above, only one study enrollment included only pyogenic liver abscesses (removal of amoebas, fungi, etc.) and concluded that PNA was superior to PCD, this is consistent with our conclusion. The findings of these other studies conflict with the findings from this previous RCT and our study. This might because these other studies included patients with multiple types of abscesses (amoebic, pyogenic, mixed, and indeterminate). The choice between PNA and PCD in the management of liver abscesses is still controversial, and as you say it is essential to carry out multi-center study, and we emphasized this in the limitations section (Lines 276-278). 

Even though this study was a retrospective study, we still found several important findings from it. The percutaneous treatment showed good therapeutic effects on PLA. Presence of positive blood culture, liver cirrhosis, and the length of fever resolution were independent predictors of primary treatment failure. Based on our experience, PNA has advantages over PCD, with higher success rates, lower costs, shorter hospital stays, a

---

## [Decision Letter · Decision Letter 1]

26 Nov 2024

Management and prognostic prediction of pyogenic liver abscess in a Chinese tertiary hospital: percutaneous needle aspiration vs catheter drainage

PONE-D-24-30205R1

Dear Dr. Gu,

We’re pleased to inform you that your manuscript has been judged scientifically suitable for publication and will be formally accepted for publication once it meets all outstanding technical requirements.

Kind regards,

Zhaoqing Du, Ph.D

Academic Editor

PLOS ONE

Additional Editor Comments (optional):

Reviewers' comments:

Reviewer's Responses to Questions

**Comments to the Author**

1. If the authors have adequately addressed your comments raised in a previous round of review and you feel that this manuscript is now acceptable for publication, you may indicate that here to bypass the “Comments to the Author” section, enter your conflict of interest statement in the “Confidential to Editor” section, and submit your "Accept" recommendation.

Reviewer #2: All comments have been addressed

2. Is the manuscript technically sound, and do the data support the conclusions?

Reviewer #2: Yes

3. Has the statistical analysis been performed appropriately and rigorously? 

Reviewer #2: Yes

4. Have the authors made all data underlying the findings in their manuscript fully available?

Reviewer #2: Yes

5. Is the manuscript presented in an intelligible fashion and written in standard English?

Reviewer #2: Yes

6. Review Comments to the Author

Reviewer #2: This study presents a comprehensive analysis of the management and prognostic prediction of pyogenic liver abscess (PLA) in a Chinese tertiary hospital, comparing two primary interventional methods: percutaneous needle aspiration (PNA) and catheter drainage (CD). Utilizing a retrospective cohort design, the research encompasses a substantial sample size, ensuring robust statistical analysis and reliable conclusions.

The study contributes to the existing body of knowledge on PLA management by providing evidence-based recommendations for interventional choices and highlighting areas for future research. The findings have implications for clinical practice, potentially improving patient outcomes and reducing healthcare costs.

7. PLOS authors have the option to publish the peer review history of their article (what does this mean?). If published, this will include your full peer review and any attached files.

Reviewer #2: No

---

## [Editor Report · Acceptance letter]

5 Dec 2024

PONE-D-24-30205R1 

PLOS ONE

Dear Dr. Gu, 

I'm pleased to inform you that your manuscript has been deemed suitable for publication in PLOS ONE. Congratulations! Your manuscript is now being handed over to our production team.

Kind regards, 

on behalf of

Dr. Zhaoqing Du 

Academic Editor

PLOS ONE